# Cardiac Involvement in Chronic Lymphocytic Leukemia/Small Lymphocytic Lymphoma

**DOI:** 10.3390/jcm11236983

**Published:** 2022-11-26

**Authors:** Tadeusz Robak, Jarosław D. Kasprzak, Dorota Jesionek-Kupnicka, Cezary Chudobiński, Paweł Robak

**Affiliations:** 1Department of Hematology, Medical University of Lodz, 93-510 Lodz, Poland; 2Department of General Hematology, Copernicus Memorial Hospital, 93-510 Lodz, Poland; 3Department of Cardiology, Medical University of Lodz, 92-213 Lodz, Poland; 4Department of Pathology, Chair of Oncology, Medical University of Lodz, 90-419 Lodz, Poland; 5Department of Radiology, Copernicus Memorial Hospital, 93-510 Lodz, Poland; 6Department of Oncohematology, Copernicus Memorial Hospital, 93-510 Lodz, Poland; 7Department of Experimental Hematology, Medical University of Lodz, 93-510 Lodz, Poland

**Keywords:** cardiac involvement, CLL, heart, ibrutinib, Richter syndrome, Richter transformation, rituximab, SLL, venetoclax

## Abstract

Cardiac involvement of chronic lymphocytic leukemia/small lymphocytic lymphoma (CLL/SLL) is recognized extremely rarely. In addition, most CLL/SLL patients with heart infiltration are asymptomatic. In this review, we present the results of a literature search for English language articles concerning CLL/SLL or Richter transformation with symptomatic cardiac involvement. In total, 18 well-described cases with CLL/SLL and heart infiltration were identified. Only three patients were not diagnosed with CLL/SLL before the cardiac manifestation. In other patients, cardiac CLL/SLL was diagnosed between 5 months and 20 years from CLL/SLL diagnosis. All patients in these series had a diagnosis of secondary cardiac CLL/SLL. In addition, we identified four reported cases with Richter transformation in the heart. The treatment of patients with CLL/SLL and cardiac infiltration is variable and depends on the previous history and clinical characteristics of heart infiltration. In addition, no recommendations exist on how to treat patients with CLL/SLL and cardiac involvement.

## 1. Introduction

Chronic lymphocytic leukemia (CLL) is characterized by the clonal expansion of mature, CD5 positive B lymphocytes in the blood, marrow, lymphoid tissue and occasionally in other organs [1,2]. CLL is regarded as similar to small lymphocytic lymphoma (SLL) by the World Health Organization (WHO) because they share several clinical features including immunophenotype (CD5+/CD19+ and CD5+/CD23+) and morphology; however, they differ with regard to the site of B-cell proliferation [3]. SLL/CLL is classified as a combined lymphoproliferative disease [2,3]. It is the most common type of leukemia in the Western world with an incidence of 3.5–5 cases per 100,000 inhabitants per year [4]. The incidence of SLL is very low in patients categorized with SLL/CLL, as about 10% of patients only demonstrate nodal involvement. The mean age of a patient with SLL/CLL is 70 years, and median survival is 10 years [1]. However, the clinical course is extremely variable, with survival time ranging from months to decades. Approximately 2–10% of patients with CLL/SLL develop an aggressive lymphoma, most commonly diffuse large B-cell lymphoma (DLBCL), known as Richter’s transformation (RT) [5,6]. The minority of patients present with symptoms at diagnosis, including constitutional symptoms, a rapid increase in peripheral blood (PB) lymphocytosis and symptomatic lymphoid tissue enlargement [7].

The present study describes the results of a search for English language articles concerning CLL/SLL, cardiac symptoms, heart involvement and Richter transformation in PubMed and Google. Publications from January 1970 to August 2022 were scrutinized. Additional relevant publications were obtained by reviewing the references from the chosen articles. The search yielded 17 well-described cases with CLL/SLL and heart involvement. In addition, four patients with RT in the heart were identified and presented.

## 2. Cardiac Involvement in Lymphoma

The liver, lungs, skin, central nervous system, gastrointestinal tract, lungs, bone, prostate and heart are occasionally involved in CLL in addition to bone marrow (BM) and lymphoid tissue [8,9,10,11,12,13]. Schwatz and Shamsuddin identified leukemic infiltrates in spleen and lymph nodes in histologic sections in 47 patients with CLL [8]. The liver was involved in 98 per cent of cases, kidney in 90 per cent, adrenal gland in 71 per cent and heart in 64 per cent. In another postmortem series, cardiac infiltration by CLL and other lymphoproliferative disorders was observed in 20% of patients [12]. However, a clinical diagnosis of leukemic heart infiltration is extremely rare. In most cases, the presenting symptoms are heart failure, arrhythmia and cardiac arrest [14]. The most common types of pathological primary cardiac lymphoma are DLBCL, followed by Burkitt’s lymphoma, T cell lymphoma, small lymphocyte lymphoma and plasmablastic lymphoma [8,14], while the most common secondary cardiac lymphomas are DLBCL, T-lymphoblastic lymphoma and Hodgkin’s lymphoma (HL) [12,15,16]. Cardiac involvement of CLL/SLL is observed extremely rarely [13,14,15,16]. In a retrospective analysis of 94 patients with heart infiltration of non-Hodgkin’s lymphoma (NHL), Gordon et al. found CLL/SLL involvement in only six cases (7%) [13]. Congestive heart failure was observed as the presenting symptom in four cases, arrhythmia in one case and cardiac arrest in one patient. Three patients were treated with chemotherapy and three were not treated. However, cardiac involvement is commonly seen in autopsies. In a postmortem study of 47 CLL patients, 25 (64%) patients had myocardial involvement, while 14 had infiltration within the epicardium, 18 within the myocardium and 12 within the endocardium [8].

Zhao et al. identified 37 cases in a Chinese population with cardiac lymphoma [16]. The cardiac manifestations included chest tightness, shortness of breath, increased heart rates and electrocardiographic abnormality caused by pericardial effusion. Lymphoid infiltrations were mainly localized in the right atrium and were larger than 5 cm. Most patients had B-cell lymphoma (n = 23) and 14 NK or T-cell lymphoma. The pathological types of lymphomas derived from B cells included one case of grade 2 follicular lymphoma (FL), three cases of FL-transformed large B-cell lymphoma, two cases of Burkitt‘s lymphoma, nine cases of non-specific DLBCL, six cases of primary mediastinal DLBCL and two cases of high-grade B cell lymphoma. No CLL/SLL cases were included in this series. However, CLL/SLL is a rarer disease in China than in Europe and North America. In this study, all cases were accompanied by pericardial effusion diagnosed via echocardiography, suggesting that tests may be of value in patients with lymphoma.

## 3. CLL/SLL Infiltration of the Pericardium

Pericardial invasion and constrictive pericarditis are unusual initial presentations for CLL/SLL (Table 1) [17,18,19]. Most of the patients had no evidence of aggressive transformation of the CLL at the time of cardiac symptom development. Habboush et al. reported the case of a 55-year-old patient with CLL complicated by constrictive pericarditis [17]. The leukemic pericardial involvement was confirmed in postmortem histology. Monomorphic infiltration of the pericardium by small lymphocytes, consistent with CLL/SLL, were documented. Danilova et al. [18] describe the case of a 71-year-old man with fatigue, progressive dyspnea, weight gain and anasarca. The complete blood count (CBC) was normal. Imaging studies revealed bilateral pleural and pericardial effusion with thickened pericardium and constrictive heart physiology. The patient underwent a total pericardiectomy. Histologic evaluation of pericardial tissue revealed fibrous pericarditis and infiltrate with CD5+, CD19+ and CD23+ small lymphocytes. The BM biopsy showed similar B-cell infiltration consistent with CLL/SLL. The patient received fludarabine and rituximab after total pericardiectomy but died from complications related to infection [18]. However, no residual CLL/SLL in postmortem examination was found. The third patient presented with symptoms of constrictive pericarditis improved breathing and exercise tolerance after pericardiectomy and was subsequently treated with bendamustin and rituximab (BR) [19]. Upon diagnosis of cardiac involvement, CT imaging of the chest revealed thickening of the pericardium and multiple anterior mediastinal and precarinal lymph nodes. Echocardiography revealed thickened pericardium and signs of pericardial constrictive physiology. In cardiac magnetic resonance imaging, abnormal septal motion and pericardial tethering were found. Radical pericardiectomy was performed with improved breathing and exercise tolerance. Histopathology revealed chronic pericarditis with fibrosis and involvement by B-cell CLL/SLL. Bendamustine and rituximab (BR) immunochemotherapy was initiated, and the patient has remained asymptomatic for more than one year at the time of publication.

Lin et al. observed a 58-year-old woman with cardiac tamponade mimicking tuberculous pericarditis as the initial presentation of CLL [20]. She was presented with dyspnea and tachycardia. Cardiomegaly and a small left pleural effusion were found in chest X-ray. CT scan revealed mediastinal lymphadenopathy and a large pericardial effusion. Transthoracic echocardiogram confirmed the presence of a large, circumferential pericardial effusion with signs of right ventricular collapse. Pericardiocentesis was performed and approximately 1L of sanguineous fluid was extracted from the pericardial sac. Histological examination of the pericardium identified infiltration of the pericardial tissue with CD5 and CD20 positive B-cell lymphocytes consistent with CLL/SLL diagnosis. The patient remained free from symptomatic disease without any chemotherapy. At one-year follow-up, no pericardial effusion recurrence was observed.

Giannini et al. present the case of a 78-year-old patient with untreated CLL who died in cardiogenic shock after brain surgery [21]. Postmortem evaluation showed pericardial effusion with 1000 mL of serosanguineous fluid. The epi- and pericardium had diffuse hemorrhages and infiltrates of CD20+ and CD5+ CLL cells. Nnaoma et al. report the case of a 61-year-old woman with SLL and pericardial effusion [22]. Pericardiocentesis was performed to prevent cardiac tamponade. Analysis of the pericardial fluid was consistent with an SLL diagnosis. She was treated with BR with a favorable outcome. Another patient with previously diagnosed asymptomatic CLL and subsequent cardiac tamponade is described by Samara et al. [23]. The patient presented with progressive respiratory failure due to evolving cardiac tamponade. Pericardiocentesis was performed, and 1150 mL of serosanguineous pericardial fluid was removed with immediate improvement and resolution of tachypnea. Morris et al. observed a case of a 68-year-old woman who developed pleuritic chest pain and pericardial effusion three years after CLL diagnosis [24]. Computed tomography (CT) imaging identified a large pericardial effusion with lymphadenopathy. Pericardiocentesis and catheter drainage was performed and a chylous effusion was completely evacuated. In the pericardial fluid, the cell count was 18.4 × 109/L with 98% lymphocytes and CLL cell immunophenotype CD19+, CD20+, CD5+, CD10−, CD38+, CD43+, CD71−, CD23+ in 32% of the cells. However, an asymptomatic pericardial effusion subsequently recurred, and the patient received six cycles of chlorambucil and obinutuzumab, resulting in complete disappearance of the pericardial effusion without second pericardiocentesis [24]. Finally, Almeda et al. discuss the case of a 64-year-old man diagnosed with a tumor infiltrating the myocardium and pericardium, which produced necrotic tissue that masqueraded as pericardial tamponade [25]. Chemotherapy with high-dose cyclophosphamide, vincristine, adriamycin and dexamethasone was introduced, but the patient died with *Aspergillus* infection.

## 4. CLL/SLL Patients with Lymphoma Heart Infiltration in the Myocardium and Endocardium

Other reported CLL/SL patients had lymphoma heart infiltration in the myocardium and endocardium (Table 2). Four cases were characterized by leukemic involvement of the mitral or aortic valve and myocardium by CLL/SLL lymphocytes [26,27,28,29]. Applefeld et al. report the case of endocardial fibroelastosis and intractable congestive heart failure caused by CLL [26]. At autopsy, endocardial fibroelastosis of the left side of the heart was identified, with marked endo- and myocardial infiltration by the CLL cells. Meltzer et al. describe the case of a 48-year-old man who developed cardiac symptoms secondary to mitral valvular dysfunction, three years from CLL diagnosis [27]. At autopsy, focal fibrosis associated with dense infiltration of uniform lymphocytes in the left atrium and ventricle were identified. The patient received a prosthetic valve as replacement. However, he died soon after recovery from the operation. Another patient with CLL/SLL and involvement of the aortic valve was reported by Chiste et al. [28]. A 77-year-old man with a medical history of CLL and chest pain revealed severe aortic stenosis and moderate mitral regurgitation in the imaging studies. The lymphocytic infiltrate specimen obtained during aortic valve replacement and mitral repair was found to be positive for CD20, PAX5, CD5 and CD23. Another CLL patient with aortic valve infiltration was presented recently by Posch et al. [29]. The patient started treatment with ibrutinib and subsequently underwent two-vessel coronary bypass surgery and aortic valve replacement. The histochemical examination found lymphocytes positive for CD20, BCL2, CD5 and CD23, consistent with CLL of the valve.

The presence of acute coronary syndrome, secondary to cardiac infiltration and coronary occlusion of CLL cells, has been recently reported in four patients (Table 2) [30,31,32,33]. Bennet et al. present the case of a patient with a history of CLL who developed clinical and ECG symptoms of coronary ischemia [30]. An echocardiogram revealed an external echogenic mass invading the anterolateral left ventricular wall. Imaging with cardiac magnetic resonance imaging (MRI) and CT revealed external encasement of left circumflex coronary artery with mediastinal mass, leading into downstream myocardial ischemia and subsequent necrosis. The patient died seven days later, before CLL treatment was initiated [30]. Assiri et al. describe the case of an 83-year-old man with CLL who died following an acute myocardial infarct; autopsy revealed infiltration of the coronary artery walls by CLL cells [31]. In addition, a recent study by Betting and Kemp [32] describes the case of a 59-year-old man with CLL, who died suddenly due to acute coronary syndrome. Again, microscopic evaluation during autopsy revealed a thick rim of lymphocytes in the adventitia of the coronary arteries and focal prominent clusters of lymphocytes in the atherosclerotic plaques. Htet et al. [33] report the case of a patient presenting with acute coronary symptoms and a troponin rise. Magnetic resonance imaging (MRI) and echocardiogram revealed full thickness infarction and hypokinesia secondary to a rising peripheral lymphocyte count in peripheral blood. The patient had no prior cardiac history and a normal coronary angiography, suggesting that the progress of CLL played a pathophysiological role in the development of coronary disease. Excellent clinical response was achieved after immunochemotherapy with BR and decreasing of lymphocytosis.

Finally, we observed a case of CLL/ SLL presenting with progressive fatigue, dyspnea and arrhythmia due to SLL cardiac infiltration, developed 20 years after diagnosis of CLL/SLL (Figure 1 and Figure 2) [34]. Immunotherapy with bendamustine and rituximab and then venetoclax and rituximab significantly reduced lymphoma infiltration of the heart (Figure 3). However, during the treatment, Richter transformation was recognized in the bone marrow; therapy with rituximab, cyclophosphamide, doxorubicin, vincristine and prednisone (R-CHOP) was introduced and CR achieved.

## 5. Richter Transformation in the Heart

RS transformation most frequently involves the lymph nodes (60–70% of cases). However, extranodal localizations have also been reported in the gastrointestinal tract (10%), tonsil (10%), BM (10%) and other sites. Finally, RT in the heart has been reported in four patients (Table 3) [35,36,37,38]. Xu et al. describe a 65-year-old woman who was diagnosed with CLL several years before RS transformation and death [35]. A complete autopsy showed diffuse lymphoid infiltration of mediastinal, para-aortic, axillary, portal, inguinal and supraclavicular lymph nodes. Histological examination found the tumor to be consistent with DLBCL. Intimal IgM lambda paraprotein deposition in myocardial arteries was found, and this was believed to be responsible for the acute myocardial infarction and sudden death. Histological evaluation of the heart revealed lymphomatous infiltrates in the myocardium, seen in the epicardium and endocardium. In addition, occluded vessels were present in the areas of acute infarction, with hypereosinophilia of myocytes and a sparse neutrophilic inflammatory reaction. Isolated cardiac RS was reported in a 61-year-old man, four years after diagnosis of stable, asymptomatic CLL [36]. A chest CT scan revealed the presence of a 5 cm cardiac mass in the right auricle, infiltrating the superior vena cava. PET-CT showed intense fluorodeoxyglucose (FDG) uptake by the cardiac mass.

Pathologic examination of the transvenous intracardiac area showed a massive lymphoid infiltration and DLBCL associated with SLL lesions. The patient was treated with four cycles of R-CHOP, and CR, confirmed in PET-CT scan, was achieved. Marra et al. describe a 69-year-old woman with tachycardia and dyspnea, and a voluminous rounded hypodense mass (41 × 58 mm in size) revealed by cardiac ultrasound and CT scan. The tumor had invaded the right atrium and demonstrated intense FDG uptake on PET scan; no other FDG-positive pathological lesions were found outside the heart tissue [37]. The patient underwent a right atriotomy with tumor mass resection and complete removal of the interatrial septum. Histological examination of the tumor showed diffuse infiltration of large pleomorphic neoplastic B cells consistent with a diagnosis of DLBCL. Bone marrow biopsy showed monomorphic small neoplastic B lymphocytes (CD5+, CD20+, CD23+), indicating the coexistence of SLL, clonally related to DLBCL. Taken together, the clinicopathological picture was consistent with Richter’s transformation in the heart. The patient was treated with R-CHOP and was alive and in complete remission at 42 months after diagnosis. Another case of RS with isolated cardiac involvement diagnosed by 2-[18F]FDG PET/CT scan was reported recently [38].

## 6. Discussion

Clinical manifestations of lymphoma localized in the heart are exceedingly rare events: cardiac lymphoma represents only 1.3% of primary cardiac malignancies, and only 16% to 28% of patients with lymphoma are localized in the heart [39]. In most patients, cardiac involvement by lymphoma is due to secondary hematogenous spread from other sites, as in our patient [40]. In hematologic malignancies, neoplastic cells can involve all components of the heart including the pericardium, myocardium and endocardium [41]. In a retrospective analysis of 94 patients with cardiac involvement of non-Hodgkin’s lymphoma (NHL) from 1990 to 2015, Gordon et al. found six cases (7%) of CLL/SLL [8].

Our literature search revealed 18 well-described cases with CLL/SLL cardiac infiltration and an additional four cases with RT in the heart. Only 3 of the 18 (17%) patients were not diagnosed with CLL/SLL before the cardiac manifestation. Cardiac CLL/SLL was diagnosed between 5 months and 20 years from CLL/SLL diagnosis. Primary cardiac lymphomas have been reported in the literature, but almost all were aggressive B-cell lymphomas [42]. Among the 37 cases with cardiac lymphoma identified by Zhao et al., only one patient had primary cardiac lymphoma [16].

The symptoms presented by lymphoma patients with cardiac involvement were highly heterogeneous; however, the most common forms included involvement of pericardium with pericardial infiltration, as identified in nine (50%) patients (Table 1). In other studies, the clinical manifestations of heart lymphoma infiltration included constrictive pericarditis, pericardial effusion, heart failure, arrhythmia and cardiac arrest [43,44,45]. However, cardiac lymphoma is easily missed or misdiagnosed and frequently requires multimodality cardiac imaging, including echocardiography and cardiac magnetic resonance imaging (MRI) [14,16,18]. Although echocardiography is the initial diagnostic test for pericardial effusion, MRI provides more detail for characterizing pericardium infiltration [22]. In addition, PET-CT can be useful for diagnosing and evaluating the treatment of lymphomas with cardiac changes, especially in aggressive lymphomas [45,46].

Pericardial involvement upon autopsy was identified in up to 20% of patients with malignant disease [47,48]. In addition, several case reports presenting with pericardial effusion have been published in the literature, including patients with NHL and Hodgkin’s lymphoma [14,15,16,17,18,49,50,51,52,53]. However, tamponade is uncommon as an initial manifestation of neoplastic disease [54], occurring mainly in lung cancer (60%), with only 9% of cases originating with leukemia or lymphoma [50]. Pericardial effusion was noted in 7 of the 18 patients with CLL/SLL and heart involvement presented herein. It was also identified in our patient under CT imaging but did not have any clinical significance. Effusion can be reduced by pericardiocentesis with an indwelling catheter, and this can prevent cardiac tamponade, as shown in two patients in our series. In addition, as shown in our patients and other patients with NHL, subsequent systemic chemotherapy reduces the reoccurrence of the pericardial effusion and reduces patient morbidity and mortality [18,21].

Among the 18 patients reported herein, 2 demonstrated acute myocardial infarction and sudden death due to coronary attack, with infiltration of the coronary vessels by CLL/SLL cells confirmed by autopsy. In one patient, cardiac symptoms developed 10 years from CLL diagnosis, and in the other, SLL was diagnosed post mortem. Infiltration of the coronary vessels by neoplastic cells have been occasionally reported in other hematologic malignancies, including primary NHL and multiple myeloma [55,56]. A high lymphocyte count in PB can play a pathophysiological role in the development of coronary symptoms [32]. Patients with CLL may also develop amyloid infiltration of the heart, in which the deposits within the vessels induce coronary occlusion [57,58].

Infiltration of the heart valves in lymphoma is very rare and occurs in most patients by direct extension from extravalvular cardiac lesions [59,60,61,62,63,64]. Our search revealed three cases of CLL cardiac valve infiltration, with the mitral valve involved in one case and aortic valve infiltration in the other two patients. In all three cases, CLL/SLL was previously diagnosed. In the case with mitral involvement, valve replacement was performed; however, the patient died soon after recovery from the operation. In the other two, replacement of the aortic valve was performed: one was reported to be doing well on ibrutinib, while the postoperative history was not reported in the other.

The treatment of patients with NHL with cardiac involvement is variable and depends on the previous history and clinical characteristics of heart infiltration. Among our identified patients with CLL/SLL, 7 out of 18 (39%) patients had pericardiectomy, pericardiocentesis or catheter drainage, with subsequent immunochemotherapy in four of them. In total, chemotherapy was used in eight (44%) of the patients. One patient was treated with ibrutinib and one with venetoclax and rituximab; however, the prognosis for these patients was poor. Only two patients survived longer than one year after immunochemotherapy, and one patient was reported to be doing well on ibrutinib at the time of publication.

Patients with NHL and cardiac involvement demonstrate variable survival. In a series of 94 patients reported by Gordon et al. [14], subsequent therapy was known only for half of them, and survival data were reported for 56. In total, 34% of the patients were treated with chemotherapy; these had longer median survival (18 months) than those who were not treated (median one month). Among patients with B-cell lymphoma, median survival was four months, i.e., longer than for T-cell lymphoma (median two months). Among patients with CLL/SLL, median survival was 37.5 months, which was longer than among those with aggressive B-cell lymphoma (median four months). In addition, patients with B-cell NHL with primary cardiac involvement lived longer (median six months) than those with secondary cardiac lymphoma (median two months).

We identified four cases with RT in the heart [35,36,37,38]. Richter syndrome is the relatively uncommon development of an aggressive DLBCL or HL in patients with a previous or concomitant diagnosis of CLL/SLL [7]. However, RT has a poorer prognosis than de novo DLBCL or HL. In most (80%) cases, RT is clonally related to CLL/SLL, and has worse prognosis than cases with clonally unrelated RT, with a median overall survival of 8–16 months. Among the cases reported here, RT in the heart was diagnosed simultaneously with SLL/CLL in one patient but developed several years from the SLL/SLL diagnosis in three others. In all patients, RT was isolated to the heart. One patient was alive and in CR at 42 months after diagnosis. Other patients responded well to immunochemotherapy, but observation only lasted a few months.

The development of sudden cardiac symptoms in patients with CLL indicates the possible presence of RT involving the heart, and proper diagnostic procedures and treatment should be initiated. Cardiac ultrasound, chest CT and a PET-CT can be useful diagnostic tools; however, the gold standard for RT diagnosis is histological evaluation with an open biopsy. In our patient, histological diagnosis of the lymphoid infiltration in the heart confirmed CLL/SLL, but not the suspected RT.

## 7. Summary and Conclusions

Heart infiltration in CLL/SLL is extremely rare: in the English language literature, only 18 cases have been reported in detail. All patients had a diagnosis of secondary cardiac CLL/SLL. Among the patients reported herein, pericardial invasion and constrictive pericarditis were observed in eight patients. Two patients demonstrated acute myocardial infarction and sudden death due to coronary attack. Our search also revealed three cases of CLL cardiac valve infiltration. In addition, we identified four reported cases with isolated cardiac RT. Cardiac lymphoma is easily missed or misdiagnosed and frequently requires multimodality cardiac imaging, including echocardiography and cardiac MRI, CT scan and PET imaging. The treatment of patients with CLL/SLL with cardiac involvement is variable and depends on the previous history and clinical characteristics of heart infiltration.

## Figures and Tables

**Figure 1 jcm-11-06983-f001:**
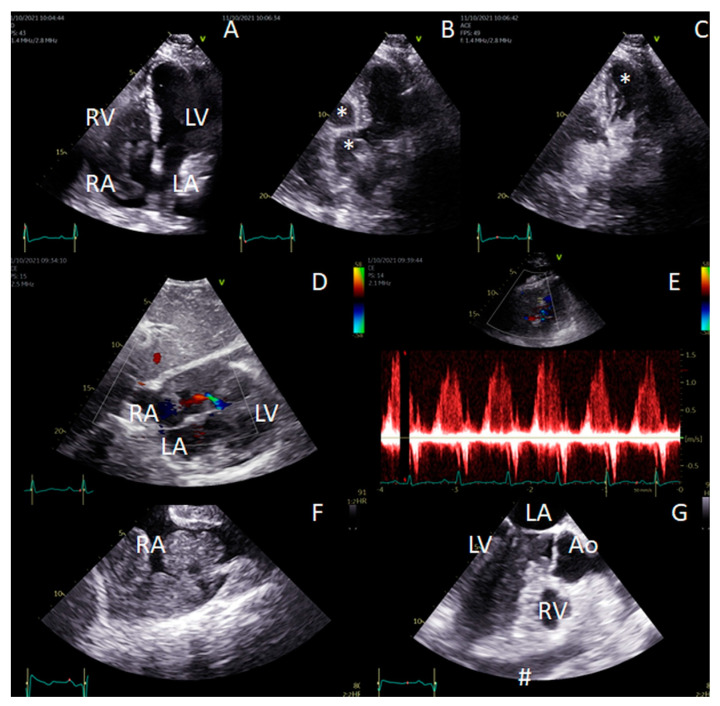
Echocardiographic features of cardiac involvement in a lymphoma patient. Images from transthoracic (TTE–(**A**–**E**)) and transesophageal (TEE-(**F**,**G**)) echocardiography. (**A**–**C**). Transthoracic echocardiogram (apical four chamber view, the same image orientation) of lymphoma cardiac masses filling cardiac cavities in non-enhanced images (**A**) and contrast-enhanced images (**B**,**C**). Lymphoma masses presenting as deficient contrast areas within cardiac cavities (in RA, RV–(**B**), early contrast inflow phase and in LV–(**C**), delayed contrast inflow phase). (**D**,**E**)–TTE subcostal view showing intracardiac masses with accelerated, turbulent color flow within inflow portion of the right ventricle consistent with inflow obstruction, confirmed by spectral Doppler (E, inflow accelerated to 1.5 m/s). (**F**–**G**). Transesophageal imaging of intracardiac masses—multiple oval tumors visible in the right ventricle and atrium, with thickened interatrial septum ((**F**)–bicaval midesophageal view) as well as within the left ventricle (LV–(**G**), 3-chamber esophageal view). Mild pericardial effusion (#) is present with epicardial tissue deposits. Abbreviations: *—lymphoma masses shown in contrast enhanced imaging as a deficient contrast area within cardiac cavity; #—pericardial effusion, Ao—ascending aorta, LA—left atrium, LV—left ventricle, RA—right atrium, RV—right ventricle.

**Figure 2 jcm-11-06983-f002:**
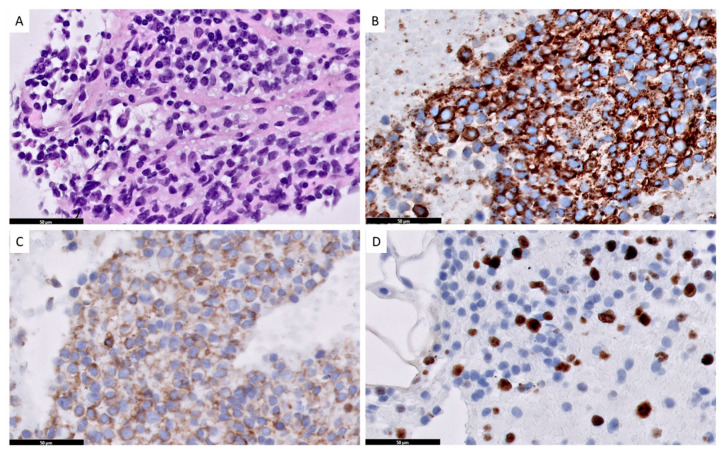
Infiltration of CLL/SLL in intracardiac biopsy of the masses in right. ventricular epicardium with cytomorphology off small B-cells (**A**) (HE × 400), immunophenotype showed strong positivity of CD23 (**B**) and CD5 (**C**) (anti -CD23 and CD5, DAKO × 400), the proliferation index Ki-67 was about 40% (**D**) (anti- Ki-67, DAKO, × 400).

**Figure 3 jcm-11-06983-f003:**
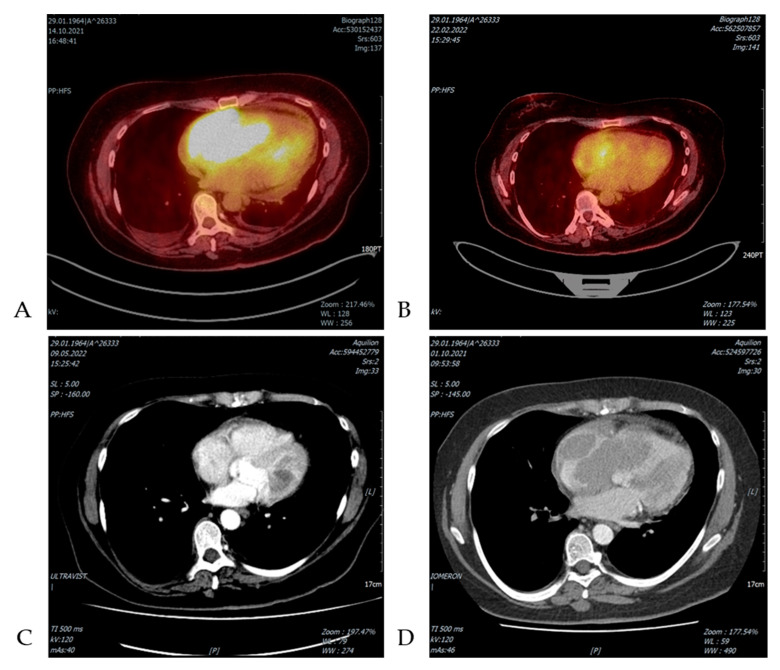
The CLL/SLL heart infiltration, located mainly in the region of the right atrium, partially right and left ventricle. The FDG PET CT scan presents a large pathological area 10 × 6 × 7 cm with high SUVmax 10.4 before treatment (**A**) and residual lesion 5 × 3 × 2 cm with SUVmax 7.5 after treatment (**B**). The analogous CT scan with iv contrast before (**C**) and after treatment (**D**) also indicates significant reduction of SLL/CLL mass.

**Table 1 jcm-11-06983-t001:** Reported CLL/SLL patients with CLL/SLL infiltration of the pericardium.

Authors	Age/Sex	CLL/SLL Characteristics	Cardiac Manifestation	Treatment after Cardiac Diagnosis	Response to Treatment
Habboush et al., 1996 [17]	55/F	Asymptomatic CLL for 5 months	Thickened pericardium over the right ventricle, at postmortem thickened and hemorrhagic pericardium with fibrinous exudate, histologically marked CLL/SLL involvement of the pericardium	Anterior pericardiectomy via a median sternotomy	At the post-operative period, development of supraventricular tachycardia, hypotension and cardiac arrest.
Danilova et al. 2015 [18]	71/M	No previous CLL/SLL diagnosis, CBC normal	Bilateral pleural effusions, pericardial effusion with thickened pericardium, and constrictive heart physiology; histopathology—CLL/SLL in pericardium and BM, imaging studies revealed ascites, and scattered lymphadenopathy	Total pericardiectomy, subsequent treatment with FR.	Patient died from infection complication, no residual CLL/SLL in postmortem examination.
Ho et al. 2018 [19]	57/M	No previous CLL/SLL diagnosis, CBC normal	CLL/SLL involvement of the pericardium, symptoms of constrictive pericarditis, CT imaging–thickening of the pericardium with pericardial effusion multiple anterior mediastinal and precarinal lymph nodes.	Radical pericardiectomy with subsequent BR treatment	Improved breathing and exercise tolerance after pericardiectomy; free of symptoms >1 year postoperatively and BR therapy
Lin et al. 2010 [20]	58/W	CLL/SLL diagnosis, simultaneous with cardiac involvement	Cardiomegaly and a small left pleural effusion, mediastinal lymphadenopathy and a large pericardial effusion in CT, with evidence of right ventricular collapse, the thickened pericardium between 0.1 and 0.3 cm.	Pericardiocentesis, pericardial involvement by CLL cells, no chemotherapy	Free from symptomatic disease at one year follow-up
Giannini et al. 1997 [21]	78/M	CLL (stage Rai 0), no treatment	Nausea and breathlessness after brain surgery, one year after CLL diagnosis, postmortem pericardium contained 1000 mL of serosanguineous fluid with CLL/SLL cells, generalized lymphoadenopathy, hepato-splenomegaly, and infiltrates of kidneys, lungs and liver	No specific treatment initiated	The patient died in cardiogenic shock.
Nnaoma et al. 2019 [22]	61/W	No previous CLL/SLL diagnosis	Pericardial effusion with a compressed right atrium, pericardial fluid and BM biopsy consistent with a diagnosis of SLL/SLL.	Pericardiocentesis followed by BR with G-CSF	Successful outcome following percutaneous pericardiocentesis and immunochemotherapy.
Samara et al. 2007 [23]	73/M	CLL Rai 0, no treatment	Tachypnea, ECG—sinus tachycardia 5 years from CLL diagnosis, moderate cardiomegaly with small pleural effusions in chest radiograph, TTE—large pericardial effusion.	Leukapheresis—initial clinical response, further management was initiated as an outpatient	No echocardiographic recurrence of pericardial fluid.
Morris et al. 2019 [24]	65/W	Previous CLL diagnosis treated with fludarabine	Pleuritic chest pain, pericardial effusion with progressive lymphadenopathy, 3 years after CLL diagnosis, pericardial fluid with CLL cells, progression of prior maxillary, mediastinal, and hilar adenopathy, with mild focal consolidation at the left lung base.	Complete evacuation of pericardial effusion by catheter drainage, 6 cycles of Chl + Obi	Complete disappearance of the pericardial effusion. No recurrence of the pericardial effusion 3 years after chemotherapy.
Almeda et al. 2001 [25]	64/M	Previous CLL diagnosis	4-month history of dyspnea, malignant replacement of the pericardium and epicardium with necrotic tissue, with diffuse infiltration of B-cell immunoblastic lymphoma	Pericardiocentesis, chemotherapy with high-dose cyclophosphamide, vincristine, adriamycin, and dexamethasone.	Death with *Aspergillus* infection during chemotherapy

Abbreviations: BM—bone marrow involvement, BR—bendamustine + rituximab, CBC—complete blood count, CLL—chronic lymphocytic leukemia, CT—computed tomography, Chl—chlorambucil, Obi—obinutuzumab, CT—computed tomography scan, ECG—electrocardiogram, FR—fludarabine + rituximab, G-CSF—granulocyte colony stimulating factor, M—male, SLL—small lymphocytic lymphoma, TTE—transthoracic echocardiogram, W—women, WBC—white blood count.

**Table 2 jcm-11-06983-t002:** Reported CLL/SLL patients with lymphoma heart infiltration in the myocardium and endocardium.

Authors	Age/Sex	CLL/SLL Characteristics	Cardiac Manifestation	Treatment after Cardiac Diagnosis	Response to Treatment
Applefeld et al. 1980 [26]	42/M	CLL diagnosed 1 8 months before cardiac symptoms, splenic irradiation	Congestive heart failure postmortem examination showed endocardial fibroelastosis and leukemic infiltration of the endocardium, myocardium, and coronary arteries	No specific treatment,	Sudden death
Meltzer et al. 1975 [27]	48/M	Asymptomatic CLL, no treatment	Cardiac symptoms 3 years from CLL diagnosis: mild congestive heart failure, secondary to mitral valvular dysfunction, cardiac catheterization—severe mitral regurgitation	Mitral valve replacement, no further treatment, infiltration by CLL/SLL cells in the surgically excised mitral valve	Death soon after recovery from operation. At autopsy dense infiltration of CLL/SLL cells in left atrium and ventricle.
Chisté et al. 2013 [28]	77/M	History of CLL/SLL diagnosis	Worsening chest pain over 8 weeks, severe aortic stenosis and moderate mitral regurgitation, CLL/SLL infiltration in aortic valve, calcification and fibrosis	Aortic valve replacement and mitral repair	Postoperative history not reported
Posch et al. 2021 [29]	75/M	CLL/SLL treated with ibrutinib	Diffuse re-stenosis of the stents and notable aortic stenosis, CLL/SLL infiltration of aortic valve, 2 years from CLL diagnosis	Replacement of aortic valve	Ibrutinib continuation
Bennett et al. 2020 [30]	72/M	Diagnosis of CLL and autoimmune hemolytic anemia	Echogenic mass surrounding anterolateral left ventricular epicardial space, infiltrating myocardium, complete occlusion of marginal coronary artery, mediastinal mass, leading into downstream myocardial ischemia and subsequent necrosis	The patient managed conservatively due to reduced general health	Palliative care, patient died a month after admission.
Assiri 2005 [31]	83/M	CLL diagnosed 10 years before cardiac manifestation	Acute myocardial infarction, postmortem examination showed endocardial fibroelastosis and leukemic infiltration of the endocardium, myocardium, and coronary arteries	Resuscitation	The patient died despite resuscitation.
Betting and Kemp 2021 [32]	59/M	No previous CLL/SLL diagnosis	At autopsy, coronary artery with severe infiltration with CLL/SLL cells in the adventis and media of the vessels and within intimal plaques, CLL infiltration in lymph nodes and spleen	No specific treatment	Sudden death due to coronary attack
Htet et al. 2019 [33]	55/W	Stable asymptomatic SLL	Acute coronary syndrome with hypokinesis in the midinterventricular septum without significant stenosis or thrombus within the coronary arteries 6 years after SLL diagnosis CLL/SLL infiltration in renal biopsy, and bone (extensive infiltration of humeral head and ribs)	Six cycles of RB every four weeks	Excellent hematologic and clinical response
Robak et al. 2022 [34]	57/W	SLL diagnosed 20 yrs prior to cardiac infiltration	Increasing fatigue and tachycardia, Echocardiogram—multiple ovoid, partly mobile intracardiac masses up to 45 mm size, identified in all cardiac cavities, the CT image demonstrated adenopathy in the mediastinum, enlargement of the cardiac silhouette and hypodense areas in the right atrium, right ventricle and left ventricle. Heart biopsy and BM trephine biopsy indicated a diagnosis of SLL.	RB, venetoclax + rituximab, R-CHOP	Excellent cardiac response after BR and venetoclax plus R with RT in BM

Abbreviations: BM—bone marrow involvement, BR—bendamustine + rituximab, CLL—chronic lymphocytic leukemia, CT—computed tomography, M—male, R—rituximab, CHOP—cyclophosphamide, doxorubicin, vincristine and prednisone, RT—Richter transformation, SLL—small lymphocytic lymphoma, W—women, WBC—white blood count.

**Table 3 jcm-11-06983-t003:** Reported patients with Richter transformation and heart infiltration.

Authors	Age/Sex	Patient Characteristics	Cardiac Manifestation	Treatment after RT Diagnosis	Response to Treatment
Xu et al. 2011 [35]	65/W	CLL several years before RT	Lymphomatous infiltrates in the myocardium, epicardium and endocardium consistent with DLBCL, massive lymphomatous involvement of mediastinal, para-aortic, axillary, portal, inguinal and supraclavicular lymph nodes.	No specific treatment	Acute myocardial infarction and sudden death
Zdrenghea et al. 2017 [36]	61/M	Asymptomatic CLL for 4 years	Cardiac mass (8 × 5-cm) in the right auricle, infiltrating the superior vena cava in CT, in PET-CT) intense FDG uptake of the cardiac mass, large right pleural effusion	1 cycle of COP and 4 cycles of R-CHOP + ASCT	CR 3 m after ASCT
Marra et al. 2021 [37]	69/W	PB normal, SLL in BM biopsy	Rounded hypodense mass (41 × 58 mm) invaded the right atrium and the interatrial septum revealed by cardiac ultrasound and PET/CT scan	Right atriotomy with tumor mass resection + 6 R-CHOP	Alive and in complete remission at 42 months
Pudis et al. 2021 [38]	75/M	CLL/SLL diagnosed before RT symptoms	Isolated cardiac involvement, PET/CT scan revealed a large cardiac mass in the right atria with high metabolic activity. Biopsy confirmed the diagnosis of DLBCL.	Not reported	Not reported

Abbreviations: ASCT—autologous stem cell transplantation, BM—bone marrow involvement, CLL—chronic lymphocytic leukemia, CHOP—cyclophosphamide, doxorubicin, vincristine and prednisone, COP—cyclophosphamide, vincristine and prednisone, CT—computed tomography scan, DLBCL—diffuse large B-cell lymphoma, FDG—^18^F-fluorodeoxyglucose, M—male, PB—peripheral blood, R—rituximab, PET—positron emission tomography, R—rituxima, RT—Richter transformation, SLL—small lymphocytic lymphoma, W—women.

## Data Availability

All patient data are available from the corresponding author (T.R.).

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
