# Peer review of "Cardiac Involvement in Chronic Lymphocytic Leukemia/Small Lymphocytic Lymphoma"

_jcm, 2022, doi:10.3390/jcm11236983_

Round 1
Reviewer 1 Report
The manuscript is clear and well written. The topic is interesting and well covered by the authors.
I believe that every paragraph is well written. Many articles have been cited correctly and completely.
I believe this review is important and appealing in this area (LLC). Cardiac involvement by CLL is rare and the findings in this manuscript are valuable.
the paper represents an interesting review of the literature on this topic. I believe it is very relevant since there is no collection in the literature on cardiac involvement in chronic lymphocytic leukemia. I believe the methodology is correct. a minimal revision of the English language is useful. the bibliography is correct and exhaustive. the tables are explanatory and the figures interesting.
Author Response
Reviewer 1
The manuscript is clear and well written. The topic is interesting and well covered by the authors.
I believe that every paragraph is well written. Many articles have been cited correctly and completely.
I believe this review is important and appealing in this area (LLC). Cardiac involvement by CLL is rare and the findings in this manuscript are valuable.
The paper represents an interesting review of the literature on this topic. I believe it is very relevant since there is no collection in the literature on cardiac involvement in chronic lymphocytic leukemia. I believe the methodology is correct. A minimal revision of the English language is useful. The bibliography is correct and exhaustive. The tables are explanatory and the figures interesting.
Authors response: We thank the Reviewer for positive review of our paper.

Reviewer 2 Report
The authors present a review article highlighting the manifestations of cardiac involvement by chronic lymphocytic leukemia/small lymphocytic lymphoma (CLL/SLL).
Table 1, 2 and 3. The cardiac manifestation column should specifically show were the CLL/SLL infiltrate is in the heart if possible. Most of the cases have the clinical manifestations but information about the location of the infiltrate is not consistent.
Make sure all references are in the article. Reference 5 and 47 appear to be absent.
Author Response
Reviewer 2.
Comments and Suggestions for Authors
The authors present a review article highlighting the manifestations of cardiac involvement by chronic lymphocytic leukemia/small lymphocytic lymphoma (CLL/SLL).
Table 1, 2 and 3. The cardiac manifestation column should specifically show were the CLL/SLL infiltrate is in the heart if possible. Most of the cases have the clinical manifestations but information about the location of the infiltrate is not consistent.
Authors response: We have re-edited Table 1, 2 and 3. Including location of the infiltration whenever data are available
Make sure all references are in the article. Reference 5 and 47 appear to be absent.
Authors response: We hacked the references an included ref 5 and 47 in the article.
